# Scalable Output Linear Actuators, a Novel Design Concept Using Shape Memory Alloy Wires Driven by Fluid Temperature [†]

**Andres Osorio Salazar** *[iD], **Yusuke Sugahara** [iD], **Daisuke Matsuura** [iD] and **Yukio Takeda** [iD]

School of Engineering, Tokyo Institute of Technology, Tokyo 152-8552, Japan; sugahara.y.ab@m.titech.ac.jp (Y.S.); matsuura.d.aa@m.titech.ac.jp (D.M.); takeda.y.aa@m.titech.ac.jp (Y.T.)

* Correspondence: osorio.a.aa@m.titech.ac.jp; Tel.: +81-03-5734-3557

† Osorio Salazar A., Sugahara Y., Matsuura D., Takeda Y. (2021) A Novel, Scalable Shape Memory Alloy Actuator Controlled by Fluid Temperature. In: Niola V., Gasparetto A. (eds) Advances in Italian Mechanism Science. IFToMM ITALY 2020. Mechanisms and Machine Science, vol 91. Springer, Cham.

**Abstract:** In this paper, the concept of scalability for actuators is introduced and explored, which is the capability to freely change the output characteristics on demand: displacement and force for a linear actuator, angular position and torque for a rotational actuator. This change can either be used to obtain power improvement (with a constant scale factor), or to improve the usability of a robotic system according to variable conditions (with a variable scale factor). Some advantages of a scalable design include the ability to adapt to changing environments, variable resolution of step size, ability to produce designs that are adequate for restricted spaces or that require strict energy efficiency, and intrinsically safe systems. Current approaches for scalability in actuators have shortcomings: the method to achieve scalability is complex, so obtaining a variable scaling factor is challenging, or they cannot scale both output characteristics simultaneously. Shape Memory Alloy (SMA) wire-based actuators can overcome these limitations, because its two output characteristics, displacement and force, are physically independent from each other. In this paper we present a novel design concept for linear scalable actuators that overcome SMA design and scalability limitations by using a variable number of SMA wires mechanically in parallel, immersed in a liquid that transmits heat from a separate heat source (wet activation). In this concept, more wires increase the maximum attainable force, and longer wires increase the maximum displacement. Prototypes with different number of SMA wires were constructed and tested in isometric experiments to determine force vs. temperature behavior and time response. The heat-transmitting liquid was either static or flowing using pumps. Scalability was achieved with a simple method in all tested prototypes with a linear correlation of maximum force to number of SMA wires. Flowing heat transmission achieved higher actuation bandwidth.

**Keywords:** Shape Memory Alloy; linear actuator; scalability

## 1. Introduction

Ever since automated systems were introduced, the ability to respond to a change, to be adaptable, was highly sought after by designers and users. Indeed, if their systems could continue to be useful by meeting the ever-changing demands from the market, it would provide value for a longer time-frame. Nowadays, although flexible automation consists of various combinations of technology, this demand for flexibility was mainly achieved through the "programmability of the controlling computers" [1]. With re-programmable systems, a robot can change the order or type of operations, in order to improve its range of application. By having flexibility, whenever there is a change in the output requirement, e.g., a change of the object being manipulated, the same robotic system can continue to be useful.

In that sense, the focus of companies is then to design systems that, besides meeting the high-quality-low-cost standard requirement, have also to facilitate rapid response to market

changes and consumer needs. Thus, the term responsiveness provides a specific goal for automation system to strive for. According to Koren et al. [2], "responsiveness refers to the speed at which a plant can meet changing business goals and produce new product models. It enables manufacturing systems to quickly launch new products on existing systems, and to react rapidly and cost-effectively to: market changes, including changes in product demand, product changes, including changes in current products and introduction of new products; and system failures (ongoing production despite equipment failures)."

Various methods to achieve this agility can be visualized. One such method it the one proposed by Kock et al. [3], where a robot concept that aims to increase the agility of newly set assembly lines by making it easier to switch from manual to fully automated assembly is presented. In their concept, the gap between the automation state of the line is reduced by using robots that can fit into assembly stations that can also be used by human workers, in combination with an intrinsically safe design for human collaboration. This allows the assembly line to react quick to changes in the number of parts processed per unit time, for example.

In spite of all the advantages steaming from the 'agility' of flexible automation, the ability to accommodate entirely new part types by easily modifying the system components is still limited. Therefore, according to Stecke et al., the next phase of flexible automation appears to be the development of reconfigurable systems, where "the technology will be designed for rapid adjustment of functionality, in response to new circumstances, by rearrangement or change of its components" [1]. Scalability for actuators, which achieves one such type of reconfigurable system, is introduced in this paper.

In the software engineering field, where this concept is widely used, a scalable design refers to the "ability of a system to accommodate an increasing number of elements, to process growing volumes of work, and/or to be susceptible to enlargement" [4]. By applying this idea to robotic systems, and using of both concepts, i.e., programmability and scalability, a system could be proven to posses advantages over its non-scalable counterparts.

Flexible, reconfigurable, and scalable robotic systems is not a concept exclusively for the assembly or production context, although the latter is its main driver considering the potential cost improving capabilities if a flexible manufacturing or assembly system is implemented. Another example in which scalability is desirable is in robots requiring mobility through rough terrain, such as the maintenance robot SHeRo presented by Agheli et al. [5]. Scalability of its output characteristics provide an extra layer of usability; in this case by giving it enhanced workspace area and longer stride length. This is done by varying the length of the legs, at the cost of robot stability and precision. Depending on the factors such as the terrain conditions, the required size of the working area and precision of motion, different length configurations of the legs are selected. An optimization algorithm for the length of the leg is proposed, considering both the stability criterion, the minimization of the robot's legs motion and the required precision of the operation being executed. In general, "operations requiring higher precision utilize the collapsed configuration while operations requiring a larger workspace or more maneuverability within a given stability range use the expanded configuration." This way, the robot SHeRo can be benefited from the improved precision of the collapsed configuration and the maneuverability of the expanded configuration by selecting an appropriate arrangement based on external factors, using scalability with a variable scaling factor, which in this case is linked to the length of its legs.

Modular robots are also widely used and are an example of scalable robots. For example, the modular robot presented by Ceron et al. [6], claims to be able to "adapt to new environments and change their morphology according to the task at hand", by changing its topology. Swarm robots, on the other hand, can provide scalability by changing the number of engaging swarm individuals, like in the study published by Alkilabi et al. [7], where their solution can respond to "variability in the physical characteristics of the object (i.e., object mass and size of the longest object's side) with scalable group sizes", considering limiting factors for larger groups such as the robustness and performance (based on success rate)

of the group. These examples demonstrate that one usage case of scalability is to provide extra usability by optimizing two or more variables that are inversely proportional to each other, i.e., the length of SHeRo's legs vs. its stability, the size of an object to be moved by the swarm proposed by Alkilabi et al. vs. the group's cardinality (therefore the robustness of its motion), etc.

Another method in which the advantages of scalable robotics could be obtained is with the design of scalable actuators, which have the ability to change the output characteristics, e.g., force and displacement for linear motion, or torque and angular position for rotational motion. Scalability for actuators, first introduced in a previous conference paper [8] and now expanded in this manuscript, will be explained in Section 2.1, along with concrete examples, and more detail on the techniques conventionally used to obtain scalable actuators with static or variable scaling ratios.

In addition to flexibility and increased usability, scalable systems have other advantages. By constructing a robot with the ability to change the output characteristics, it could be possible to construct prototypes with different dimensions for easier testing, and change to the full-scale version when its evaluation is completed. In addition, a scalable robotic system can be intrinsically safe if the maximum output is correctly designed and it is within the restriction of the applications or the physical construction of the robot.

Moreover, with a scalable system, a specific application can be designed according to its design variables, e.g., the resolution of the step size could be varied according to the needs of the usage case, like in the work of the robot SHeRo by Agheli et al., presented previously, an effect called henceforward "resolution variation". In Figure 1, both lines represent the output characteristics of the same system with different configurations, e.g., a pneumatic cylinder with two different diameters, config. 2 being larger than config. 1. With the same input signal, both systems would deliver a different output quantity, and the slope of the output curve would be related to the scale factor of each configuration. In a system or a situation where large output is required, config. 2 could be preferred, while config. 1 would be reserved for circumstances where higher precision is needed. This precision, defined as a measure of the smallest incremental output resolution that the actuator is able to achieve, comes from the fact that the slope of the relationship of input and output of the actuator is less pronounced, i.e., compared with config. 2, config. 1 produces a smaller output increment with the same input increment. For the minimum input step increment, e.g., the resolution of the controller, config. 1 produces a smaller incremental output, i.e., higher precision.

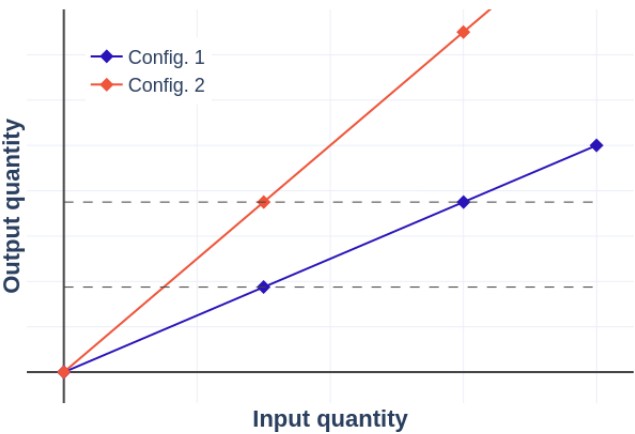

**Figure 1.** Output characteristics of a system with scaled configurations.

Finally, a scalable system can produce designs that are adequate for restricted spaces or that require strict energy efficiency. As described before, having an scalable actuator means being able to decide the design parameters precisely, in this case, setting the footprint (for a restricted space) or the precise energy output to deliver an acceptable efficiency, given that the input energy remains constant after the reconfiguration.

To take advantage of the increased usability brought by scalable actuators, the objective of this research was to construct and evaluate a design concept of an actuator that can simultaneously scale force and displacement output with minimum changes in footprint, weight, number of components, or power source requirements. The idea is that the scaling method used should be simple enough that a variable, situation dependent scale factor can be obtained.

As explained more in detail in Section 2.4, this concept consists of variable number of SMA wires mechanically in parallel immersed in a liquid used to transmit heat from a separate heat source (Wet SMA activation). This concept presents two features: more wires increase the maximum attainable force, and longer wires increase the maximum displacement. The focus of this paper was to test the first feature, force scalability, of prototypes with multiple configurations. They were tested with isometric experiments to measure force output behavior. The data obtained were used to determine the scalability performance of each prototype, given by the maximum output force relative to the quantitative construction change of each configuration, in this case, the number of wires in parallel. This variable is manipulated manually for this study, but a design that automatically modifies the number of wires in parallel will be done as a future work.

With the previously mentioned objective in mind, first, a general description of how scalability in actuators is conventionally achieved will be presented, contrasting their strong and weak points for various applications. Then, arguments for using SMA wires in a scalable linear actuator will be provided, as well as challenges for its usage with Joule heating. Next, a brief introduction of wet activation and the current state of this research field will be included, with commentary of ways in which it could be improved with the novel conceptual design presented in this paper. In the next two sections, the description of two different prototypes using the design concept presented before, with the materials and methods for its testing and discussion of the obtained results, will be detailed. Finally, the general conclusions and the planned future work will be provided.

## 2. Conceptual Design

### 2.1. Scalable Actuator Design and Its Strategies

Changing the internal or external components of an actuator, or modifying its external mechanical transmission components, can modify its output characteristics. Scalable actuators can be grouped into two categories according to their purpose:

1. To obtain power improvement, with a static scale factor.
2. To improve the usability of a robotic system according to variable conditions, with variable scale factor.

The transmission box of an automobile is a good example of a variable scale factor. In this case, the variable condition of the system is a change of torque requirement according to road conditions, or the change angular velocity of the wheels of the vehicle, therefore the velocity of vehicle itself. By automatically (or manually) changing configurations, i.e., "changing gear", different ratios of motor to wheel angular velocity can be obtained. In contrast, with a constant scale factor in a common servomotor, this ratio remains constant unless the gearbox is completely changed.

While conventionally designing for scalability with a static scale factor is trivial, even intuitive, scalability with a variable scale factor poses extra challenges, namely in the complexity of the method used to achieve such scalability, which limits the means to vary the scaling factor. By constructing an actuator in which the scaling method is done without a complex technique, this gap can be brought closer.

In general, four approaches have been identified to achieve scalability in actuators:

1. Scaling by changing reduction ratio of transmission: Changing the output quantities of an actuator by using transmission elements, e.g., the gear box of a servomotor, can achieve scalability. This is the most common method to achieve variable scaling factor. However, by using only this method, both displacement and force scalability (or

velocity and torque, in the case of the servomotor) can not be obtained simultaneously, i.e., no energy scaling.

2. Scaling of configuration, based on modular design: The output of each modular element contributes to the overall output of the actuator, like the concept presented by Britz [9] for a rotational actuator. This method could be used to obtain a variable scaling factor but the fact that most of the time it requires a complicated assembly and wiring between elements makes this approach challenging.

3. Power source scaling: Some actuators possess a range of acceptable power sources that can be related to its output energy, e.g., pneumatic cylinders: the higher the supply pressure, the larger the forces that can be generated by it. This method is also practicable for a variable scaling factor, but the range of the scalability is linked to the limitations of the actuator itself and its power source, and often times this power supply variation represents a significant cost, causes safety concerns to arise, and is in most cases difficult to achieve.

4. Energy conversion elements scaling: Done by replacing or complementing the elements that transform energy inside an actuator. Contrary to "scaling of configuration, based on modular design" that uses multiple actuator modules, this approach aims to use a single casing, control scheme, and power source. For example, if the winding of a motor is changed, its maximum output torque can be changed too, or changing diameter of a pneumatic cylinder to modify its force output response. This method mainly posses the same drawbacks as "scaling of configuration, based on modular design", since it is based on the same principle, but has the extra limitation of being restricted to the same actuator's footprint. In this case, the actuator is designed considering a maximum force, and then it can derated without a reduction in area, which is considered an undesirable characteristic. In addition, this approach is usually complicated and time consuming.

Considering scaling methods for conventional actuators that do not require to completely replace the actuator, i.e., changing the winding of a motor, or changing the size of a pneumatic cylinder, a summary of scaling methods can be found in Figure 2, each evaluated with the criterion "Simultaneous Force and Displacement scaling", i.e., energy scaling, and by the relative complexity of each solution considering the objective to achieve a scalable actuator with variable scaling factor.

| Type of actuator | Scaling method | Scaling procedure | Added / changed parts per configuration | Simultaneous Force & Displacement | Complexity score |
|---|---|---|---|---|---|
| Any | 1. Ratio of transmission | Change the input/output ratio of transmission | - | ❌ | ✅ |
| Electromagnetic actuator | 2. Modular scaling | Multiple elements in parallel & serires | Power source, relays | ✅ | ❌ |
| Pneumatic or hydraulic cylinder | 2. Modular scaling | Multiple elements in parallel & serires | Solenoid valves | ✅ | ⚠️ |
| Electromagnetic; pneumatic or hydraulic cylinder | 3. Power source scaling | Voltage & current control scheme. Pressure & flow control valve | - | ⚠️ | ✅ |
| Piezoelectric or Electrostatic | 4. Energy conversion element scaling | Change size of material or stacked arrangement | Power source | ⚠️ | ❌ |
| SMA (Joule Heating) | 4. Energy conversion element scaling | Multiple parallel elements, change length of wires | Power source | ✅ | ❌ |
| Wet SMA | 4. Energy conversion element scaling | Multiple parallel elements, change lenght of wires. Separate temperature and flow source | - | ✅ | ✅ |

**Figure 2.** Scalability evaluation of conventional actuators.

All approaches that use the method "scaling by changing reduction ratio of transmission" can not provide energy scaling. Since this approach is the simplest, it is also the most widely used for achieving a variable scaling factor as a response to variations of conditions of operation.

An electromagnetic, pneumatic, or hydraulic actuator system can be scaled with the "scaling of configuration, based on modular design" technique by adding elements in parallel for steeper force response, and in series for increased displacement; in this arrangement it is possible to achieve simultaneous force and displacement scaling. The usage of idle elements, with the aid of relays for electromagnetic and solenoid valves for cylinders, can produce the effect of "resolution variation" (Section 1) of systems with variable scaling factor, where an on-demand scalability factor can be selected, therefore they are included in the comparison in Figure 2. In the case of the electromagnetic actuator, the required power from an electrical power source increases proportional to the number of elements, which must be decided in advance to consider power restrictions in the conducting cables and/or heat dissipation considerations in the power source. In the case of the pneumatic or hydraulic systems, the required flow rate of the actuating fluid is proportional to the number of elements. This consideration does not need to be done in advance other than deciding the flow coefficient of the pressure or flow control valves of the system. For this reason, and the fact that the assembly of multiple elements in the manner described before becomes problematic with a larger number of elements, the complexity score of an electromagnetic actuator is ⊗, while the score of the pneumatic or hydraulic cylinder is ⚠.

On the other hand, using "power source scaling" for obtaining a variable scaling factor in electromagnetic, pneumatic, or hydraulic cylinders is more widely used and effective. By using pressure and flow controlling valves, or a voltage and current controlling scheme, it is possible to obtain an specific energy input to a cylinder or an electromagnetic actuator respectively; this change can be done on-demand. The range of achieved scaling, however, is restricted to the maximum specifications of the actuator itself (maximum allowable pressure for cylinders and maximum temperature, related to power dissipation in the coils for electromagnetic), which is most of the time limited. Larger variations require additional impractical considerations, such as oversized conductors and controlling elements, as well as extra safety factors. In addition, the "resolution variation" effect does not occur in this case since the resolution of the controller, which is constant, directly determines the resolution of the output. For this reason, the "simultaneous force and displacement scaling" criterion for this technique is ⚠.

Piezoelectric or electrostatic actuators can be scaled using the method "energy conversion elements scaling", by changing the size of the material or by using a stacked arrangement. Their miniaturization characteristics are excellent (relative high generated force and small displacement for piezoelectric, and an increased maximum energy density with downsizing for electrostatic). As a result of their difficulty to implement in larger applications, the "simultaneous force and displacement scaling" criterion for them is ⚠. For their requirement of complex assembly, it is very difficult to obtain a variable-scaling-factor actuator and their complexity score is ⊗.

SMA wires can also use the "energy conversion elements scaling" to provide a variable-scale-factor output. Explained with more detail in Sections 2.2–2.4, two approaches can be used as energy source: Joule heating and wet activation. The complexity score of Joule heating is ⊗, given the difficulty of supplying and controlling a low-voltage, high-current power source required for simultaneously activating multiple wires in a bundle, as will be explained in Section 2.2. Wet-activation complexity score is ✓ because the change of scaling factor is simply achieved by changing the length or number of wires mechanically in parallel. In addition, wet-activated SMA actuators, as well as hydraulic and pneumatic cylinders, can exploit the fact that their power source is distributed, and that they have intrinsically energy storage properties, i.e., high pressure in a tank for cylinder or high temperature in a reservoir for a wet-activated SMA actuator.

### 2.2. SMAs for Scalable Actuator Design and Its Challenges

Scalability using SMA wires has been seldom explored in literature, and it is mostly focused on obtaining power improvements with a constant scaling factor. An example is given by the work done by Mosley et al. [10], where multiple SMA wires are arranged mechanically in parallel to increase the force output of an actuator. A more recent example is given by Britz et al. [9], where the maximum output angle of a SMA rotational actuator is determined by the number of modules arranged in series, with the objective of being "adapted to various application specifications by customizing the rotational angle and the output torque." Both examples show the traditional conception of scalability, as a design variable that decides the permanent or semi-permanent behavior of the actuator, but can not react to different usage cases instantaneously; they require decommission and reconfiguration to be adapted. Of course, means to provide automatic variable scalability are possible to be implemented on them, but, again, the complexity of the scalability method, e.g., the difficult wiring of each single wire for the first example, and the need to add a new control circuit for each module in the second example, deems this effort complicated.

In this work we present a design concept of a scalable actuator based on SMA wires that achieve scalability by having its energy conversion elements reconfigured (Scalability strategy number 4 from Section 2.1), with a change of scaling factor that is simple. SMA was selected for this research because it has a desirable characteristic for constructing scalable linear actuators: its two output quantities, displacement and force (dependent on wire length and cross-sectional area, respectively), are physically unrelated to each other. Hence, both output quantities can be modified independently to achieve a scalable design, e.g., longer wires and multiple wires in parallel for displacement and force scaling, respectively. For this reason, an SMA linear actuator is able to independently scale displacement and force.

A wire-driven SMA linear actuator conventionally uses an electrical current passing through it, i.e., Joule Heating, as its heat source, and its output energy is obtained from the shortening of the wire and the generated tensile force. The usage of Joule heating and wires in parallel for force scaling is challenging. The components must be insulated between each other in order to avoid underpowered sections, and the power source required for parallel scaling is also taxing. As each new SMA element is added, the total resistance reduces, and the supply current requirement increases; it becomes critical in bundles with a large number of elements, and the partial solution is to arrange them mechanically in parallel but in series electrically [10]. Hence, the actuator becomes unreliable and harder to control.

Shrinking rate and fatigue performance are related to each other and they are a challenge as well. Nominally, a contraction rate of 3–4% can be achieved reliably with good fatigue performance. Some approaches have been researched: using static pulleys [11], and using springs [12]. The geometry also influences fatigue performance, as noted by Tamura [13]. SMA wires were reported to be superior to SMA springs, where the recovery force and strain of SMA springs decreased by 30% after 1000 cycles and by 60% after 10,000 cycles. In comparison, with good design practices (no overheating, not exceeding recommended fatigue strength of 350 MPa and a strain of 3–4%), SMA wires can safely perform for $10^6$ cycles [14].

Furthermore, SMA actuators struggle with actuation bandwidth due to a "relatively high heat capacity and density, they experience difficulty in transferring the heat rapidly into and out of the active element" [14]. Many approaches have been proposed in the past to improve its cooling ratio, which is the critical aspect to improve actuation bandwidth in SMA: free convection using heat sinks, forced convection [15], fluid immersion [16], and fluid heat transmission (called wet-activated SMA actuator) [12,17], the latter resulting in the best performance.

### 2.3. Wet-Activated SMA Linear Actuators

Multiple studies have been done exploring the idea of using a liquid in order to reduce the cooling cycle time. Tadesse et al. [18], Cheng et al. [19], Nizamani et al. [20], Ades et al. [21], among others, combine Joule heating with liquid cooling. These approaches show a reduction of cooling time of 20–30%, but increased heat loss also causes more power consumption [20]. While it is true that wet heating is not as fast as Joule heating, only a scalable design that uses a complex arrangement can be achieved, for the reasons explained in Section 2.2.

Expanding the previous idea, some researchers have used liquid for both heating and cooling cycles, with different ways of supplying and removing the thermal energy. Approaches include pumps [22] or pressurized vessels [17] in combination with mixing [12] or solenoid valves, and heat sources such as external heating elements or fuel [23]. In addition, the geometry of the SMA was either a straight wire [17,24] or springs [12,22].

In the work of Park et al. [12], thermal energy is introduced to the actuator using water, and its temperature control is done with a three-way valve that controls the mixing ratio between a hot and cold water reservoirs. The actuator tested in their experiment is a linear SMA actuator that uses multiple SMA springs in parallel. The springs can deliver superb maximum contraction (more than 50%), but it suffers from lower fatigue performance, as stated in Section 2.2, delivers lower output force, and can only be scaled with a complex method. In general, mixing valves like the ones used by Park, have high pressure drops and low time constants, in Park's work it was 260 ms for a full stroke, compared with the typical response time of 5–10 ms for solenoids.

The great majority of the efforts on wet SMA actuators are focused on improving the actuation bandwidth. Researches done by Flemming et al. [25] and Britz et al. [9] also consider the possibility of designing an scalable SMA actuator. The focus of the former is the development of a control method to drive an array of wet SMA actuators using valve manifolds, while the latter is the development of a rotational scalable SMA actuator that can add modules in series to amplify its output. Both of their designs can only scale through "scaling of configuration, based on modular design", which, as stated before, relies on complicated configurations, changes the actuator footprint, power source, and, in the case of Britz's design, requires scaling of the controlling electronics. This paper, however, describes the possibility of using the wet-activation technique not only for faster actuation time, but also for improved scalability, through the use of "energy conversion elements scaling", with a very simple way of changing the scaling factor in order to facilitate its variability, for the designs that are derived using the concept that will be presented in the next subsection.

### 2.4. Conceptual Design

The concept of the actuators introduced in this work can be resumed in two points:

1. SMA wires as basic actuating element: Utilization of long SMA wires as actuation units, with the advantage of higher fatigue performance, and faster actuation rate. Wire bundles allow actuators to be force-scalable when arranged in parallel, and the "static pulleys" [11], or any other approach that permits to have longer wires can be used for displacement-scalable designs.

2. Liquid thermal transmission: By using an external heat source instead of Joule heating, temperature sensing can be simplified, preventing over-stress and decreasing heat-related fatigue present in Joule heat SMA actuators [14]. It also allows the wires to be placed closer together, increasing its energy density. To transfer the heat from the external heating source to the SMA wires, a thermal conductive liquid is used, in a method called wet SMA actuator [17]. This method can overcome the problems of usage of wire bundles: insulation between elements and power supply scaling, as well as bring higher actuation bandwidth [12]. The requirements for thermal conductive liquid can be summarized in the following:

   (a) High thermal conductivity: for improved heat transfer.

(b)     High boiling temperature: for faster actuation.

(c)     Low viscosity: for better liquid displacement.

Figure 3 shows the abstraction of the concept. It can be seen that the heat source in (a) is located alongside the SMA wires, while the heat source for (b) is located outside the actuator, allowing the energy source to be obtained from different means other than electrical, e.g., solar, residual heat, etc.

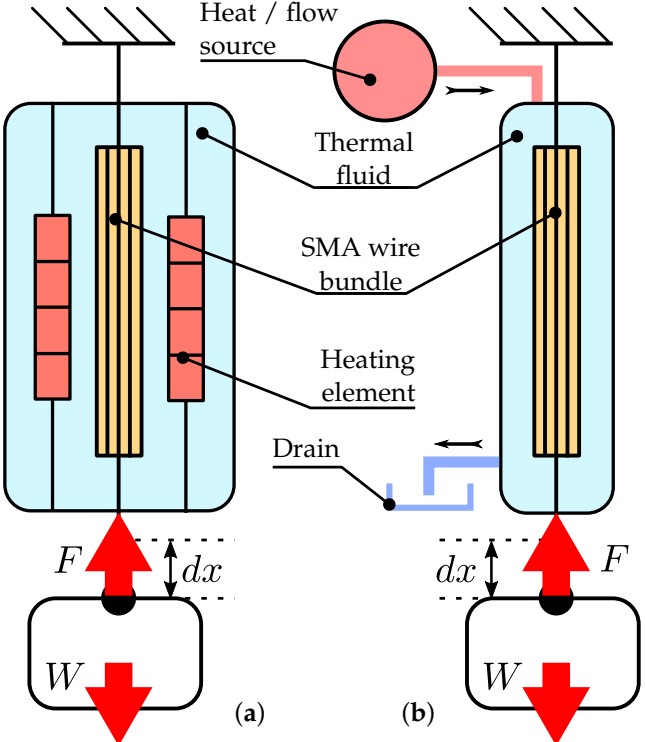

**Figure 3.** Novel concept abstraction for: (**a**) internal heat source, (**b**) external heat source.

As stated before, this concept proposes that it is possible to achieve scalability with a variable scaling factor if SMA wires are used, since the change that produces such a factor variation is simply adding more or longer cables in parallel. Automatic means for the robotic system to determine the specific configuration to be used can be implemented with a mechanism that adds or removes portions or the complete length of the wires that are activated simultaneously, for example by using a threaded rod that captures the non-required length of SMA. A proof of concept of such an arrangement will be done in a future study.

The extent of the benefits that scalability for actuators using wet SMA wires can bring to robotic systems is not limited to removing the need of a bulky transmission. In the following paragraphs, the advantages of the proposed concept will be explained by using a concrete application example. For this purpose, robots that require variable energy outputs, such as jumping or throwing robots, will be addressed, given that they benefit the most from this variable scaling factor. In their case, by being able to change the input energy of the jumping or throwing mechanism, the precise energy needed to throw or jump a required distance could be selected. This is something done usually with the method "scaling by changing reduction ratio of transmission", that uses a high transmission ratio and potential energy storage to select the energy output, a procedure that is time consuming and the output range it can provide is limited.

A common alternative is using "power source scaling" or "energy conversion elements scaling" in combination with pressurized cylinders. An example is presented by Tsukagoshi et al. [26], where a single pneumatic cylinder was used, and the energy storage was done within the robot with a pressurized tank. Increasing the supplied pressure or

flow rate ("power source scaling") were first considered as solutions for increasing the jumping height. However, since "there is a limit on the feasibility of increasing both of them, since both choices lead to increases in the whole mass of the robot", a constant scaling factor was determined using optimization techniques that maximized jumping distance in relationship with the optimal cross sectional area and mass of the components ("energy conversion elements scaling"). The relationship between the jumping height and the cross sectional area of different cylinders was also analyzed. As a result that the pressurized cylinder is on-board, its size must be minimized, which limits the storage capabilities of the robot, constraining it to only a few jump executions.

Another alternate approach, the use of the proposed concept, would give the system a higher scalability range and a decreased risk of over-pressure events that need to be mitigated in a pneumatic system with additional devices such as pressure relief valves. The advantages of SMA, i.e., high energy density and the possibility of constructing lightweight and compact systems, in combination with the benefits that the presented concept provides, i.e., energy storage, simple construction that allows easy change of energy output scaling factor, make these applications a good match for the proposed concept. In contrast with jumping robots that use pneumatic or hydraulic as a power source, i.e., a pressure source, an approach with the proposed concept uses thermal energy, with the benefit of removing the need of a separate compressor and pressurized containers for storing the energy. A simple electric heating element submerged in an integrated atmospheric reservoir would be sufficient for energy storage to allow the robot to perform an infinite number of operations while being untethered, without constrains of the number of executions for robots with on-board energy storage, and to eliminate the need for safety devices in the event of overpressurization. In addition, a higher scalability range could be achieved with the proposed concept, given that, with a pressurized power source, this range is limited to the system's maximum allowable pressure in a robot with a single cylinder, or to the flow coefficient of the components and operating pressure in a robot with multiple cylinders, whereas for SMA actuators it is only the volume of the wires activated what defines this range.

To demonstrate the scalable capabilities of the proposed concept, the next sections detail two variations: a Force-Scalable Wet Zero-Flow SMA Linear Actuator (ZFSMALA) similar to Figure 3a, and a Force and Displacement Scalable Wet Constant-Flow SMA Linear Actuator (CFSMALA) that is based on the concept shown in Figure 3b. The purpose of the first prototype was to serve as a proof of concept for evaluating if force scalability is attainable using the concept proposed. The second prototype introduced an external heat source in order to expand the application range and decrease its the response time.

## 3. Force-Scalable Zero-Flow SMA Linear Actuator (ZFSMALA)

To test the force scalability characteristics of a wet SMA wire bundle actuator, a prototype was designed, and tested [27]. This design accommodates the possibility of changing the maximum output force by adding SMA wires in parallel. It is called "Zero-Flow" because the thermal liquid is heated inside the actuator's body, i.e., there is no thermal-transmission liquid flow. As shown in Figure 4, the SMA wire bundle is assembled between two pulleys, surrounded by a heating element, submerged in a thermal fluid, and encapsulated with bellows.

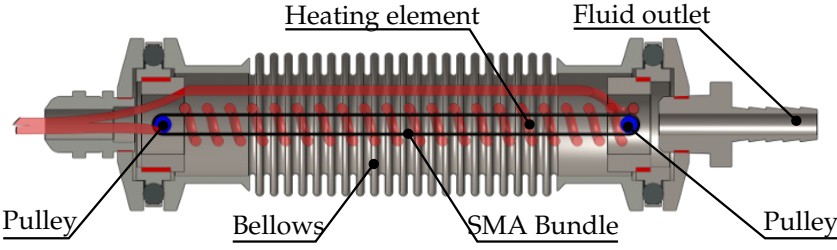

**Figure 4.** Zero-Flow Shape Memory Alloy Linear Actuator (ZFSMALA) concept.

### 3.1. Materials and Methods for the ZFSMALA Prototype

An isometric test was performed, which consists of fixing one end of the actuator to a frame and the other side to a force sensor, i.e., the actuator length being held constant. A diagram of the arrangement of the experiment can be seen in Figure 5 and the specifications of the components can be seen in Table 1. Instead of using water, as many researches described in Section 2.3 do, this experiment used an ethylene glycol aqueous solution with a boiling point of 122 °C, for increased thermal conductivity and testing range.

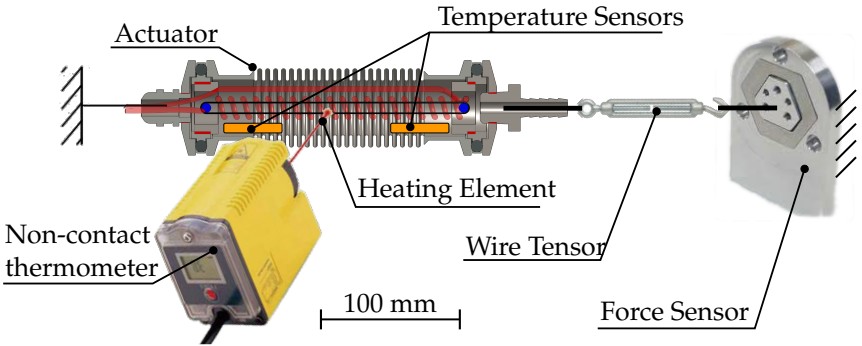

**Figure 5.** Zero-Flow experiment setup.

**Table 1.** ZFSMALA experiment components and specifications.

| Component | Parameters | | Component | Parameters | |
|---|---|---|---|---|---|
| Heat conductive liquid | Composition | Ethylene glycol: 74–76% Water: 20–22% | SMA | Diameter [µm] | 375 |
| | Boiling point [°C] | 122 | | Activation temperature [°C] | 70 |
| | Freezing point [°C] | −46 | Bellows | Material | 316 SS |
| | Thermal conductivity [J/kg·°C] | 3140 | | Spring constant [N/m] | 4.8 |
| | Viscosity [mPa·s] | 0.98 | Heating element | Power [W] | 250 |

In this experiment, the number of wires per bundle was varied from ten wires to forty, with increments of ten. The addition of bundles was done with the aid of a wire carabiner fixed to one of the pulleys. Each bundle consisted of five wires crimped together as shown in Figure 6 on each side and folded in half. Both crimped ends of each bundle were inserted on the carabiner, and the folded side placed around the pulley on the opposite side. The assembly of the bundles was done with the aid of a device equipped with two open pulleys, in order to provide a constant length for all the wires in the bundle and consistent length between all bundles.

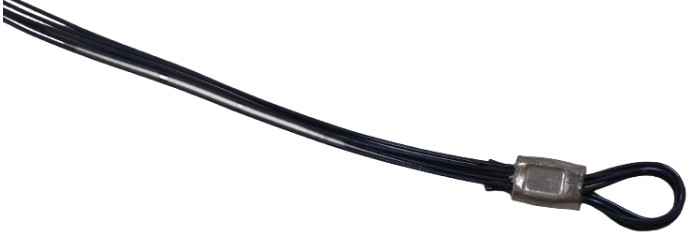

**Figure 6.** Crimping method for SMA wire bundles.

The maximum stress applied to the wires was limited to 172 MPa, according with the recommendations from the manufacturer for best fatigue performance [28]. These wires are one-way type, so an arrangement to input 35 MPa of pre-stress was provided using a turnbuckle attached to one side of the actuator. An exhaust fluid reservoir was connected to one side of the actuator to avoid any pressurization inside the chamber due to the contraction of the bellows and the heat expansion of the thermal fluid. A picture of the experiment set-up can be seen in Figure 7.

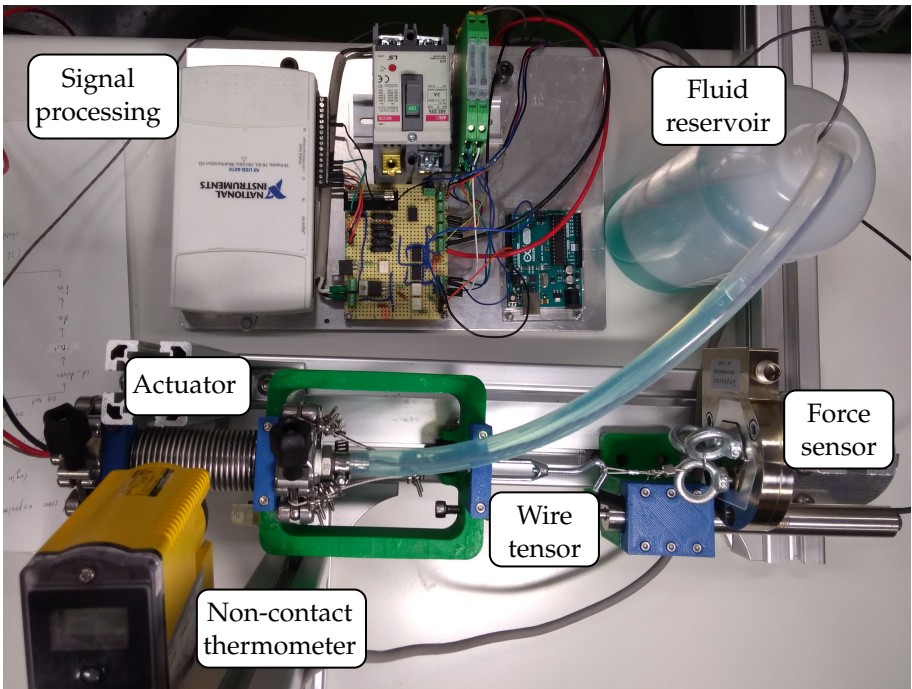

**Figure 7.** Zero-Flow experiment.

First, the pre-tension was applied with a slope of 10 N/min and sufficient time was provided in order for the tension to deform uniformly all the wires in the bundle, stopping when the tension from the force sensor registered the desired value for pre-stress constantly during five minutes. Then, the heating element located inside the bellows and next to the SMA bundles was energized, and the SMA chamber's temperature was measured on each end of the actuator with 3-conductor platinum resistance temperature detectors (RTD) and the external shell's temperature with a non-contact infrared (IR) thermometer.

The data acquisition software was self-developed using NI Labview © Version 2019, which was interfaced through a data acquisition (DAQ) device USB-6210 for the temperature sensors, with an Arduino UNO for the heating element control, and an Arduino Mega with a force sensor, as detailed in Figure 8. The test was done with a sample frequency of 10 Hz, 30 samples per cycle (averaged). No means for filtering the data were implemented for the acquisition step, although a moving-average smoothing scheme was devised to present the results. For the RTD sensors, a two-step signal conditioning was required to obtain the data: first, the RDTs were connected to a measuring transducer to obtain a 0–10 V signal, which was then passed trough a unitary amplifier to reduce the signal's impedance for the DAQ device, as recommended by NI in the technical whitepaper titled "Using a Unity Gain Buffer (Voltage Follower) with a DAQ device" [29].



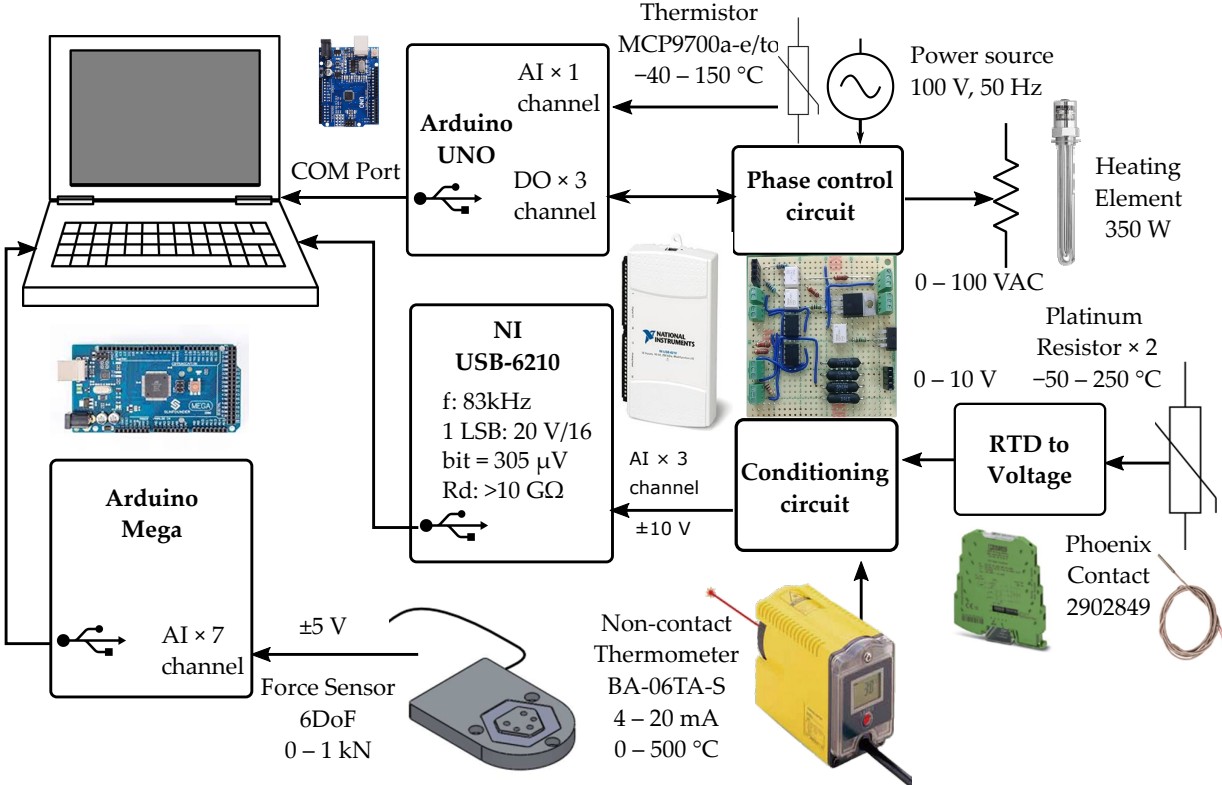

**Figure 8.** Zero-Flow (ZFSMALA) electronic block diagram.

For each SMA bundle setup, the maximum output force was determined as the intersection of the heating and cooling hysteresis curves, as can be seen in Figure 9. This point in the hysteresis cycle represents the end of the austenite transformation of the SMA material, where the output stress increment per temperature increase beyond this point is significantly smaller than during transformation, even with large temperature differences. All experiments went beyond this temperature, but only this point was considered for the evaluation of scalability. In addition, the complete heating and cooling cycles were captured in order to determine the time response of the force output of the actuator.

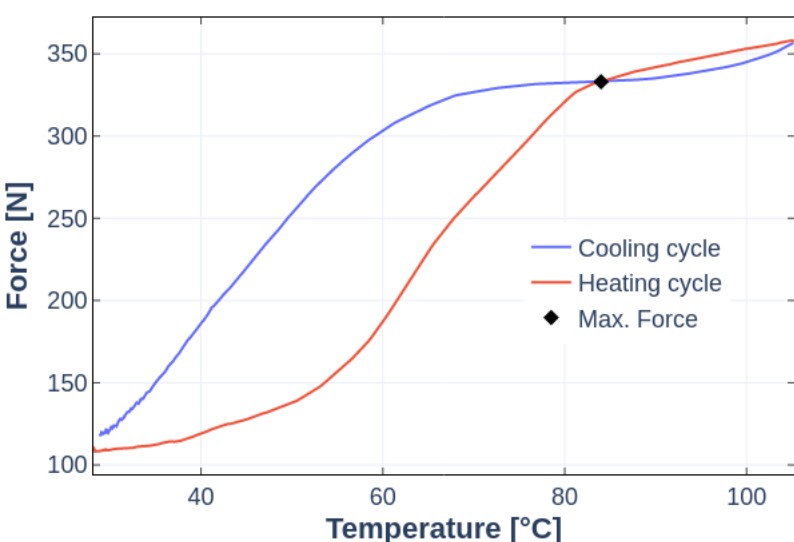

**Figure 9.** Typical force vs. temperature response for the ZFSMALA with a 30-wire bundle.

### 3.2. ZFSMALA Results and Discussion

Figure 10 shows the obtained Force vs. Temperature response of the tested configurations. Their maximum force value as described in the previous subsection are condensed in Figure 11. It can be seen that the output force of the actuator shifts up 80.2 N with each wire bundle added. It shows a direct linear correlation between the number of wires and the maximum actuating force, proving the scalability of the design concept. As described before, this variation could be achieved without any change in the power source and maintaining the same footprint. Insulation between wires was not required, and even overlapping bundles were possible.

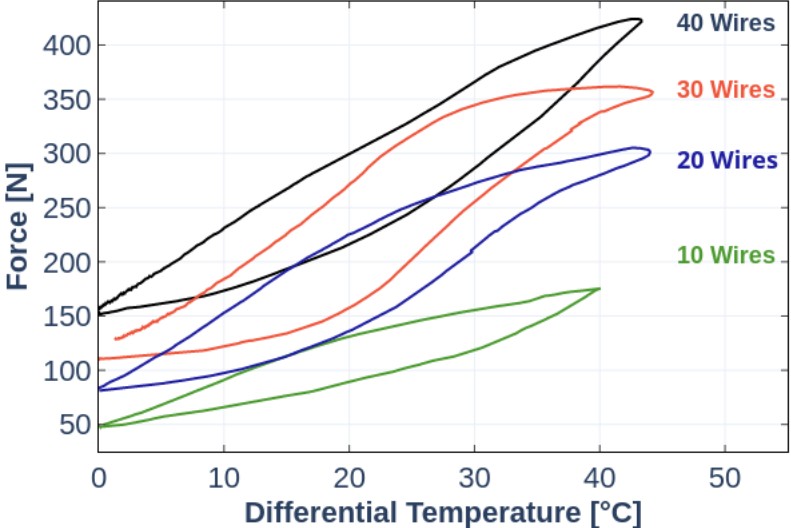

**Figure 10.** Force vs. temperature response of ZFSMALA.

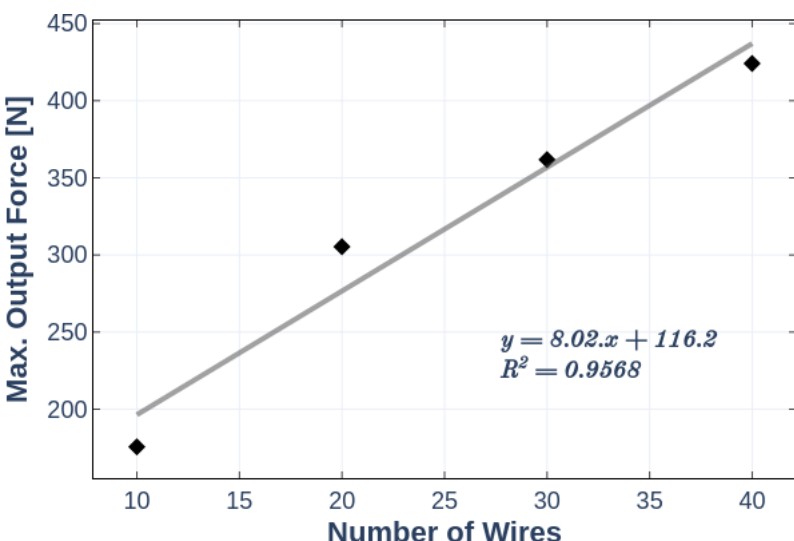

**Figure 11.** Number of wires vs. maximum force of ZFSMALA.

The time constant of this actuator was significant, especially in the cooling phase, as can be seen in Figure 12.

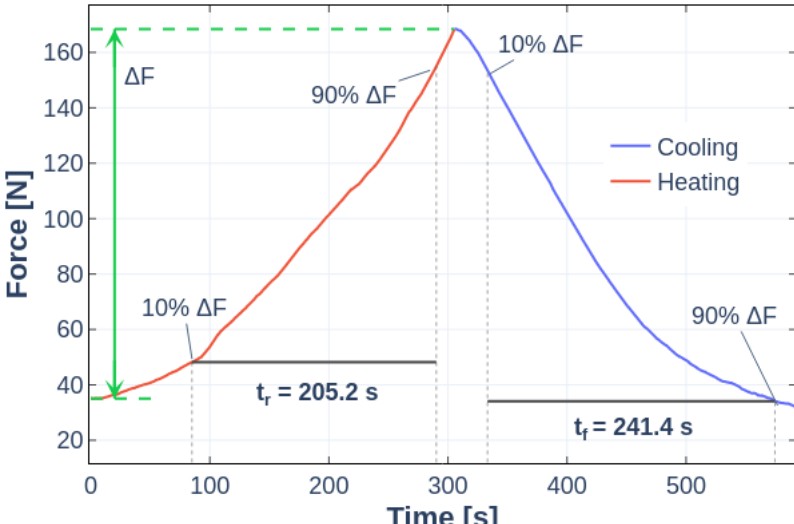

**Figure 12.** Typical force response of ZFSMALA (Bundle of 10 wires).

This time-constant is dependent on various thermal factors:

1. The volume of liquid inside the SMA chamber.
2. The heat losses trough the bellows.
3. The power output of the heating element.

A design that reduces the heat mass of the actuator, that lowers the losses through its wall, or that has a higher-power heating element could potentially achieve faster actuation. Another alternative is to consider the required heat to be prepared and stored before the actuation is used, then transporting the heat to the SMA chamber when needed. This approach will be explained in the next section.

## 4. Force and Displacement Scalable Constant-Flow SMA Linear Actuator (CFSMALA)

This experiment is called "Constant-Flow" because there is a constant flow of thermal liquid inside the SMA chamber of this actuator. This allows the heat to be prepared beforehand, and the actuation time to be dependent mainly on the liquid transportation arrangement. For this test, a force and displacement scalable linear SMA actuator was designed using the 'SMA wires, wet-activated' design concept. Its preliminary performance tests were done to verify the maximum output force of the same wire configurations as the ZFSMALA for comparison, as well as the actuation time of the heating and cooling cycles, for which fluid flow step impulses were introduced.

### 4.1. Materials and Methods for the CFSMALA Prototype

Park et al. [12] used an isometric setup for testing their wet SMA actuator, where the tested actuator is fixed between a load cell and a frame. Based on Park's work, Figure 13 shows the block diagram of the experiment setup designed in this study, and Table 2 indicates the specifications of the components. Similarly to Park's experiment, it uses a hot (preheated at 85 °C) and cold (ambient temperature 22 °C) fluid reservoirs. Contrasting with Park's design, in the present design the temperature control achieved with the mixing ratio of both reservoirs is not done using a mixing valve. Instead, the equipped pumps were selected based on the premise of controlling the individual output flow-rate of each pump, using sensorless brushless DC (BLDC) pumps and controllers. This removes the restrictions of slow actuation time and restricted flow of mixing valves.

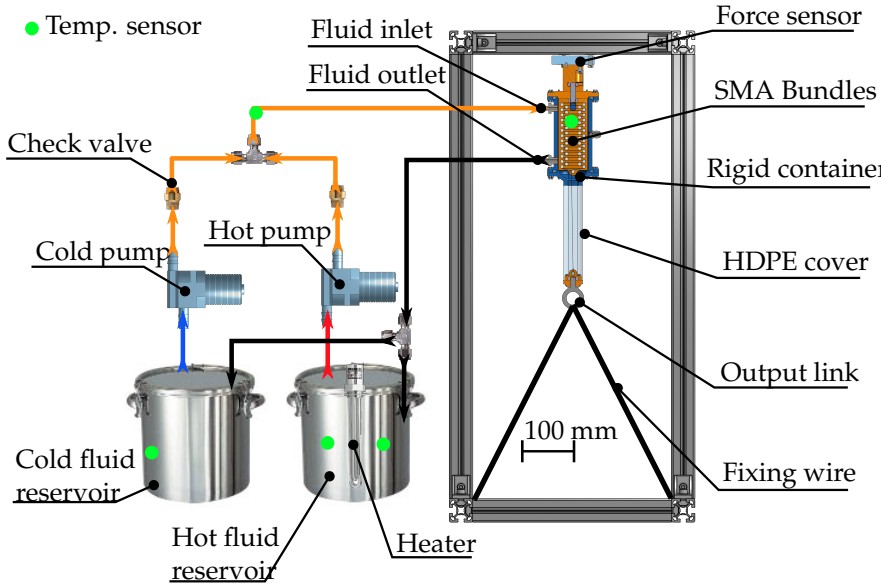

**Figure 13.** Constant-Flow (CFSMALA) experiment setup.

**Table 2.** CFSMALA experiment components and specifications.

| Component | Parameters | | Component | Parameters | |
|---|---|---|---|---|---|
| | Model | M510S-V | SMA | Diameter [µm] | 375 |
| Pump | Flow rate range [L/min] | 0–8.4 | | Activation temperature [°C] | 70 |
| | Type | Brushless | Compliant element | Material | HDPE |
| Heating element | Power [W] | 350 | | Thickness [mm] | 0.1 |
| Flow meter | Range [L/min] | 0–10 | Check Valve | Crack pressure [kPa] | 7 |

As this experiment aims to determine the actuation time, it does not require precise temperature control for now. Thus, the pumps were arranged to ideally provide an alternating flow step impulse to the actuator with a magnitude of 3.2 L/min, measured with a rotor-type mechanical flow meter. For future experiments, the temperature control will be achieved by variations of the output flow ratio of both pumps.

This design includes a set of 3D-printed carbon fiber-reinforced internal panels to freely mount static pulleys in the desired arrangement, with the idea of being able to demonstrate displacement scalability. An example of wire assembly can be seen in Figure 14. Similarly to the ZFSMALA prototype before, force scalability is to be achieved by placing bundles of wires in parallel. In addition, if the number of pulleys is increased, the length of the bundles that the actuator is capable of containing increases without changing the actuator's footprint, thus scaling displacement. A zig-zag arrangement was proposed and tested to maximize the maximum length and minimize the actuator's volume. The details of the construction of the actuator can be seen in Figure 15.

The pulley panels are placed inside a 3D-printed rigid container that creates the fluid flow channel, through fluid inlet and outlet ports. The square shape of the channel that can be seen in Figure 16, was decided in order to minimize volume inside the SMA chamber, while maximizing space for the arrangement of SMA wire bundles. The two BLDC pumps, one for hot liquid and another for cold liquid, provide the thermal-transmission liquid flow in and out the actuator. Check valves are placed in the outlet of the pumps to prevent back-flow. To complete the SMA chamber and provide a output link that allows contracting motion, a compliant High Density Polyethylene (HDPE) tube is placed between the bottom side of the rigid container and the output link of the actuator.

The compliant tube was self-manufactured using HDPE film, by sealing both sides with a manual heat sealing machine. Both ends of the tube were attached to the 3D-printed

parts with screw compression clamps, shown in Figure 16, and polytetrafluoroethylene (PTFE) film tape was used to avoid liquid leakage.

The data acquisition software was also self-developed using NI Labview © Version 2019, with the same sample frequency of 10 Hz, 30 samples per cycle (averaged), same as the ZFSMALA experiment. Similar data post-processing techniques were employed.

Firstly, the same wire bundles, wire arrangement, and attachment method from ZFSMALA were used to compare the results of both experiments, and the same pre-stress apparatus and methodology as with the ZFSMALA prototype were implemented. This arrangement was used to determine the maximum output force of each configuration and the SMA actuation time.

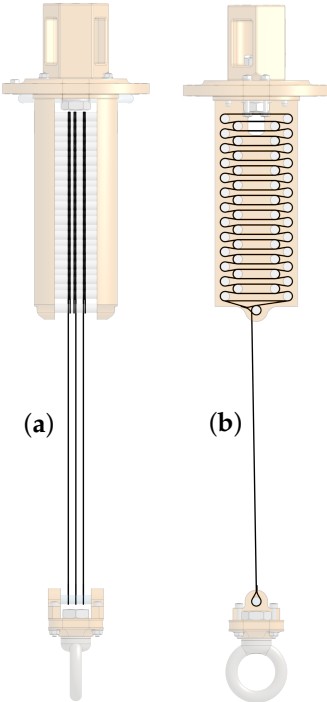

**Figure 14.** (**a**) Wires in parallel (force scaling). (**b**) Single long wire (displacement scaling).

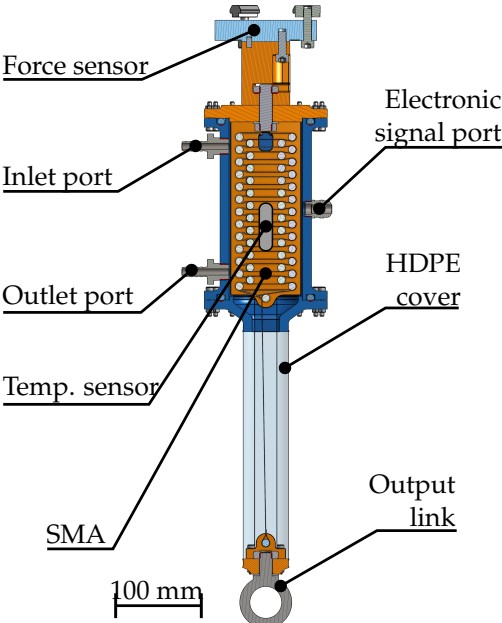

**Figure 15.** Construction details of the CFSMALA.

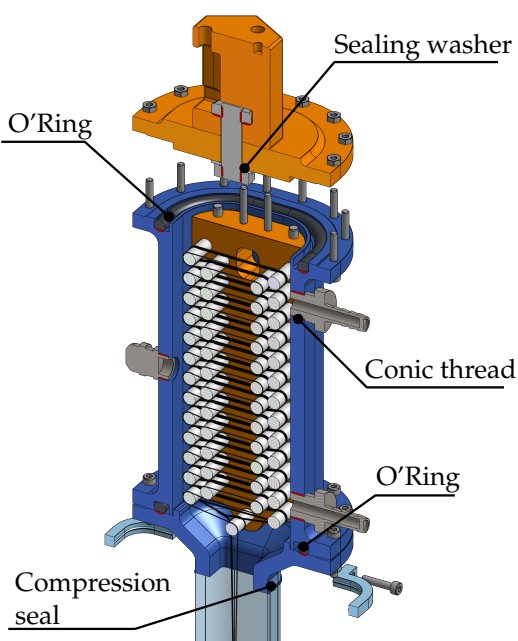

**Figure 16.** Details of the sealing of the CFSMALA prototype.

## 4.2. CFSMALA Results and Discussion

This design displays the same correlation of maximum force and number of wires corresponding to a scalable design as the ZFSMALA, as can be seen in Figure 17, with an increase of 76.34 N with each wire bundle added. The decrease in output force of the CFSMALA with respect to ZFSMALA can be attributed to the losses caused by the internal pressure in the actuator induced by the fluid flow, not present in the ZFSMALA. This decrease represents a loss in efficiency proportional to the flow rate inside the actuator, in average of 8.3% for a flow rate of 3.2 L/min. There is potential for this situation to be mitigated, for example, by placing the inlet and outlet flow ports parallel with the line of action of the actuator, by decreasing the surface area of the top and bottom faces of the actuator, or by placing a liquid flow source in the outlet of the actuator.

As can be seen in Figure 18, the rise time of this prototype from 10% to 90% is significantly faster in the CFSMALA than the ZFSMALA, 17 versus 205.2 s, respectively in the heating cycle, and 11.8 versus 241.4 s, respectively in the cooling cycle.

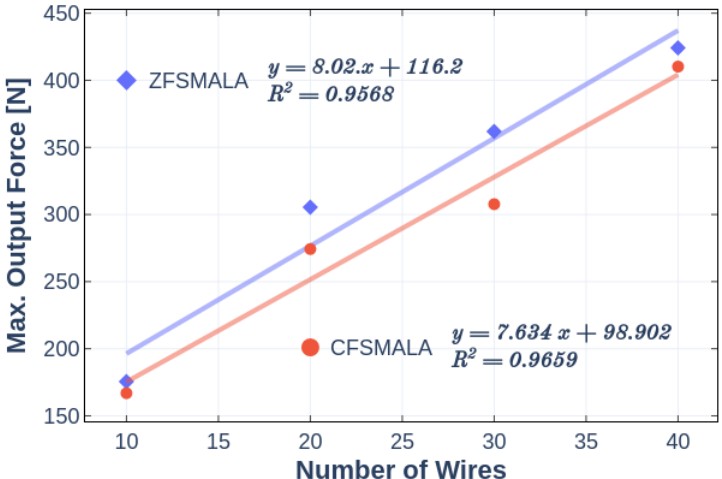

**Figure 17.** ZFSMALA and CFSMALA number of wires vs. max. force.

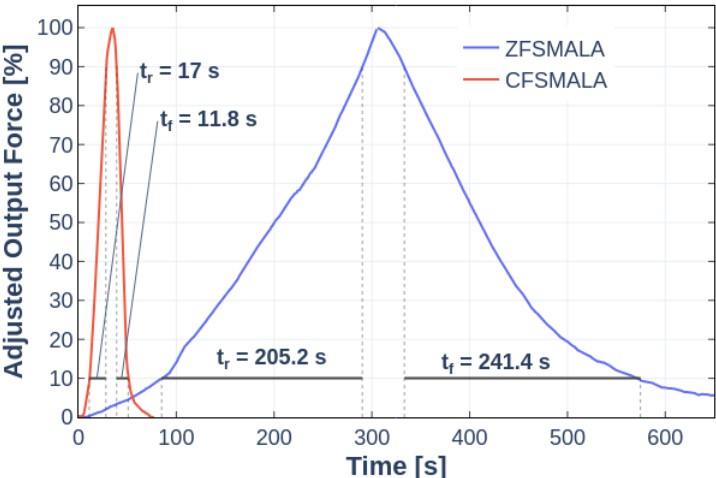

**Figure 18.** ZFSMALA and CFSMALA force response (Bundle of 10 wires).

## 5. Conclusions and Future Work

In this work, the concept of scalability for actuators was presented and detailed. A proposed division of scalability depending on its purpose, for increased power performance or for enhanced usability according to variable conditions, was given. Then, the conceptual design and general principle of operation of wet SMA scalable actuators were described. The basic performance evaluation of two scalable SMA linear actuators based on the proposed conceptual design was carried out, where the feasibility of both prototypes was tested with isometric experiments. It was found that the maximum output force of both can be linearly scaled by changing the number of actuating SMA wires placed mechanically in parallel, a change that is simply placing more (or less) wires between the pulleys. None of the actuators needed to change its size, and same heating element placed inside of the actuator for the ZFSMALA and outside for the CFSMALA was used as the power source for all the configurations of each actuator. The change of scaling factor is simple enough that automatic means of changing it could be implemented to obtain a variable scaling factor actuator, where a robot could automatically select a configuration according to its usage conditions. This change will be explored in the future, with the proposal to use variable pulley arrangements with threaded drums that release or withhold SMA sections to stop them from being activated.

The addition of forced convection heat transmission, i.e., the addition of a liquid flow to the CFSMALA, reduced the actuator's time response, and added the possibility of using a non-electrical power source, such as solar energy or residual heat. Despite the change from free convection to forced convection heat transfer, the maximum force output, thus the scalability slope of the actuator, remained constant regardless of the method for transferring heat to the actuator's chamber. For this reason, it is important to note that the proposed method for controlling the temperature inside the actuators presented is only two possibilities out of the multiple sources of energy. Heat, in the form of a free convective heat as in the ZFSMALA, or as a forced convective volumetric flow, as in the CFSMALA, is the only requirement for actuation.

None of the measuring elements in the experiments have an effect on the actuating motion. This sensor-less operation characteristic, in combination with the scalability of actuators as presented in this paper, can yield devices which can operate without an electronic controller. Instead of changing the parameters of the controller, the designer could make use of the scalability principle to change output parameters of actuators based on the presented design concept, and with the method described in this paper, e.g., changing the number of wires arranged mechanically in parallel. Consequently, this actuator can have potential applications in environmental systems where the heat power source can be the sun, large output forces are required, and slower actuation frequencies are not an issue.

The capabilities of the CFSMALA presented in this work will be expanded in the future in two ways:

1. A variable flow temperature controller to bring force/displacement control to it.
2. Wire arrangements to provide displacement scaling, with consideration of the friction losses caused by the contact of the pulleys with the SMA wires.
3. Arrangements to provide variable scaling factor to change the number of wires arranged in parallel automatically.

In addition, the development of an SMA actuated astable temperature oscillator valve will be done, whose purpose is to deliver a liquid with alternating temperature to the inlet of the actuator. An actuator like the CFSMALA would then work with a reciprocating motion, allowing to be used in increasingly more applications. There is special interest regarding sensor-less solar panel cleaning apparatus, solar tracking devices, or environmental observation robots.

Finally, while the activation time of all the presented prototypes is still too slow to be used for the application example presented in Section 2.4, jumping robots, design changes, such as the reduction of the SMA chamber's volume, usage of SMA wires with lower activation temperature or diameter, increased flow rate, usage of a different thermal conducting liquid, higher reservoir storage temperature, etc., will be explored in order to reach the required activation time. The usage of wet-activated linear SMA actuators for this application would bring certain advantages such as the capability of carrying a high load, unthetered operation, large energy storage, and precise controllable energy output.

A scalable design brings flexibility to automatic systems that can be optimized for each specific usage case and circumstance, considering design constraints such as step resolution, weight reduction, response frequency, maximum required output, and intrinsic safety.

**Author Contributions:** Conceptualization, A.O.S., Y.S., D.M., and Y.T.; data curation, A.O.S.; formal analysis, A.O.S.; funding acquisition, Y.S. and Y.T.; investigation, A.O.S.; methodology, A.O.S. and Y.S.; project administration, Y.T.; resources, Y.S.; software, A.O.S.; supervision, Y.S. and Y.T.; validation, Y.S., D.M., and Y.T.; visualization, A.O.S.; writing—original draft, A.O.S.; writing—review and editing, A.O.S., Y.S., D.M., and Y.T. All authors have read and agreed to the published version of the manuscript.

**Funding:** This work was partially supported by JSPS KAKENHI Grant Number JP17H03162, JKA and its promotional founds from KEIRIN RACE.

**Institutional Review Board Statement:** Not applicable.

**Informed Consent Statement:** Not applicable.

**Conflicts of Interest:** The authors declare no conflict of interest.

## Abbreviations

The following abbreviations are used in this manuscript:

| | |
|---|---|
| SMA | Shape Memory Alloy |
| Ni-Ti | Nickel-Titanium alloy |
| Nitinol | Nickel-Titanium alloy |
| ZFSMALA | Zero-Flow SMA Linear Actuator |
| CFSMALA | Constant-Flow SMA Linear Actuator |
| RTD | platinum Resistance Temperature Detector |
| DAQ | Data acquisition device |
| IR | Infrared |
| BLDC | Brushless DC |
| HDPE | High Density Polyethylene |
| PTFE | polytetrafluoroethylene |

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
