# Peer review of "Scalable Output Linear Actuators, a Novel Design Concept Using Shape Memory Alloy Wires Driven by Fluid Temperatureâ€"

_machines, doi:10.3390/machines9010014_

Round 1
Reviewer 1 Report
The authors describe an interesting concept for SMA wire activation, which is by fluid with controlled temperature.
This type of activation comes with several advantages in comparison to the well established Joule heating, but also has several drawbacks.
The article makes an original contribution, which although might be better suited to the scope of the Journal "Actuators".
The scientific methods are presented clearly and explained sufficiently.
The article is well written and easy to read. The length is appropriate and figures and tables are clear.
The work is motivated by e.g. agility or flexibilty in production/automation - here could be more references to existing work on especially adaptive and reconfigurable SMA systems that are developed for the same reason (e.g. grippers, robots, end-effectors...)
My only remark is that although this is a very interesting approach, the significance remains a little unclear. How is the main advantages of SMA actuators (high energy density, light weight, compact systems, cost efficient, simple design, ...) upheld with this, or why do the advantages of this proposed method outweigh? I think a concrete application example couzld help the comparison.
Thank you for this nice contribution
Reviewer 2 Report
The paper is a properly extended version of a published conference paper.
I enjoy reading the paper.
Please describe if and how a higher built-in ad-hoc electronics could improve your system.
I consider the paper valuable for Machine MDPI as it is.
Reviewer 3 Report
This paper purports to present a scalable actuator, i.e. one that can provide a range of output forces and/or displacements without a change in system size.
This is achieved by using various numbers of shape memory allow SMA threads in order to vary the force output or changing the serpentine length of the threads to adjust the displacement. This is all achieved within a single casing. A number of methods of applying the necessary heat to the SMA is presented.
The main claim of this paper seems to be that this actuator is scalable. However I do not see this system to be any different than any other in that regard. It is trivial to include parallel actuators of any type in a single casing and remove some if the generated force is not required. The system as presented is instead a system that is designed for a maximum force and can then be de-rated without any reduction in area (not a good characteristic).
While some of the implementation details are interesting, I do not understand the point of this paper and I would suggest that if I have misunderstood the point then the authors should try to explain it more clearly.
Round 2
Reviewer 3 Report
I believe the authors have addressed the reviewers' previous comments.